# ReST: Remarkably Simple Transferability Estimation

## Abstract

Existing transferability estimation methods for pre-trained neural networks suffer from method complexity, requiring extensive target data and labeled samples to predict transfer performance. We introduce ReST, a remarkably simple yet effective approach: It only requires a small subset of unlabeled samples from target data and analyzes the stable rank—a robust measure of matrix effective dimensionality—of the final layer representations. We demonstrate that this simple metric strongly correlates with transfer learning success across diverse tasks and architectures. Through comprehensive experiments on vision transformers and CNNs across multiple downstream tasks, we show that this remarkably simple approach not only matches but often exceeds the performance of sophisticated existing methods. ReST achieves 4.6% improvement over state-of-the-art methods, establishing stable rank as a powerful predictor for transferability assessment and fundamentally challenging the need for complex analysis in transfer learning evaluation. The code is made anonymously available at https://anonymous.4open.science/r/random-07C2 to ensure reproducibility of our results.

## 1 Introduction

Transfer learning has revolutionized deep learning by enabling models pre-trained on large-scale datasets to be adapted for downstream tasks Kornblith et al. (2019). However, selecting the optimal pre-trained model remains a costly trial-and-error process. With hundreds of models available—from CNNs to Vision Transformers, practitioners need efficient methods to predict transfer performance without extensive experimentation Yosinski et al. (2014); Nguyen et al. (2020b).

Current transferability estimation methods face **practical limitations**. LEEP Nguyen et al. (2020b) and LogME You et al. (2021b) require substantial labeled target data. SFDA Shao et al. (2022) and LEAD Hu et al. (2024) solve computationally expensive optimization problems. Recent methods like ETran Gholami et al. (2023b) suffer 15.5% performance degradation in label-free settings. These approaches analyze entire networks while overlooking the geometric structure of learned representations, despite evidence that feature geometry fundamentally determines transfer success Neyshabur et al. (2020); Yosinski et al. (2014).

We aim to enable **practical** model selection in resource-constrained settings by proposing **ReST**, a geometric approach that **achieves strong performance through simple computations**. Our key insight: effective transfer depends on two geometric properties captured by stable rank: a continuous measure of effective dimensionality Roy & Vetterli (2007). Models with distributed weight representations (higher stable rank) **generalize better** to new domains Sanyal et al. (2020); Neyshabur et al. (2020), while moderate activation changes between source and target indicate adaptation flexibility Yosinski et al. (2014); Wang et al. (2023).

**The method is remarkably simple and computationally efficient.** Using **a very small set of target samples**—with an effective size equivalent to only about 2 samples per class, yet without requiring class labels or exact class balance—we compute: (1) the stable rank of the classifier (c) and penultimate (p) weight matrices, and (2) the relative change in activation stable rank between source and target features at the penultimate layer. Their product yields a transferability score. **The entire computation reduces to matrix norm calculations**, requiring **no optimization and no labeled data**.

We validate ReST across comprehensive model hubs containing diverse CNN architectures (ResNet, DenseNet, MobileNet, etc.) and Vision Transformers (ViT, DINO, PVT, Swin, etc.) on 11 image classification benchmarks. ReST **achieves 4.6% higher correlation** with actual fine-tuning performance compared to the state-of-the-art methods, while operating **without any labeled target data**.

Our contributions:

- A **label-free** transferability estimator requiring **minimal unlabeled samples**
- **Computational efficiency** through **simple matrix operations** on final layers only
- **4.6% improvement** over state-of-the-art methods across diverse architectures
- Insights into what geometric properties make representations transferable

## 2    PROBLEM MOTIVATION

### 2.1    STABLE RANK AND NEURAL NETWORK GEOMETRY

The stable rank provides a differentiable measure of the effective dimensionality of neural network layers. For a matrix $W \in \mathbb{R}^{m \times n}$ with singular values $\sigma_1 \geq \sigma_2 \geq \cdots \geq \sigma_r > 0$, the stable rank is defined as:

$$\mathrm{srank}(W) = \frac{\|W\|_F^2}{\|W\|_2^2} = \frac{\sum_{i=1}^r \sigma_i^2}{\sigma_1^2} \tag{1}$$

where $\|W\|_2 = \sigma_1$ is the spectral norm and $\|W\|_F = \sqrt{\sum_i \sigma_i^2}$ is the Frobenius norm. The stable rank satisfies $1 \leq \mathrm{srank}(W) \leq \mathrm{rank}(W)$, achieving its maximum when all singular values are equal.

Unlike the algebraic rank, stable rank varies continuously with matrix perturbations and remains invariant under scaling and orthogonal transformations Roy & Vetterli (2007). These properties make it particularly suitable for analyzing neural networks during training and transfer, where weight matrices undergo continuous optimization.

### 2.2    GENERALIZATION BOUNDS AND TRANSFER LEARNING

Recent theoretical work has established tight connections between stable rank and generalization Sanyal et al. (2020). For a depth-$L$ network with weight matrices $\{W_\ell\}_{\ell=1}^L$ trained on $n$ samples, the generalization error admits the bound:

$$\mathcal{R}_{\mathrm{gen}}(\mathcal{F}) = O\left( \frac{\prod_i \|W_i\|_2}{\sqrt{n}} \sqrt{\sum_i \mathrm{srank}(W_i)} \right) \tag{2}$$

This bound exhibits an important structural property: while spectral norms contribute multiplicatively, stable ranks contribute additively under the square root. This asymmetry implies that reducing the stable rank of even a single layer can linearly improve the overall bound—a property that should be particularly valuable for the final layers most responsible for task-specific adaptation.

In the context of domain adaptation, the classical bound Ben-David et al. (2010) decomposes target error as:

$$\varepsilon_t \leq \varepsilon_s + d_{\mathcal{H}\Delta\mathcal{H}}(\mathcal{D}_s, \mathcal{D}_t) + \lambda^* \tag{3}$$

where $\varepsilon_s$ is the source error, $d_{\mathcal{H}\Delta\mathcal{H}}$ measures domain divergence, and $\lambda^*$ is the optimal joint error. This decomposition motivates our approach: we use stable rank to quantify both source generalization capacity (related to $\varepsilon_s$) and cross-domain representation shifts (related to $d_{\mathcal{H}\Delta\mathcal{H}}$).

### 2.3    REPRESENTATIONAL CHANGE AND ADAPTATION FLEXIBILITY

A fundamental question in transfer learning concerns how much representational change is beneficial versus harmful. We approach this by examining activation matrices $H_d^\ell$ for domains $d \in \{s, t\}$ using small samples from each. The quantity $\mathrm{srank}(H_d^\ell)$ captures the effective dimensionality of representations at layer $\ell$, and we can track domain-specific changes through representational shifts.

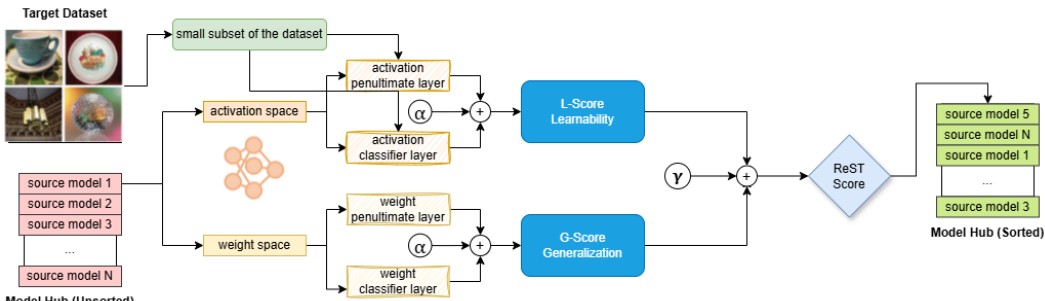

Figure 1: Overview of the ReST framework for geometric transferability estimation. Given a target dataset and a model hub of pre-trained source models, ReST evaluates transferability through two geometric components: (1) **G-Score (Generalization)**: analyzes stable rank in weight space of penultimate (p) and classifier (c) layers to measure intrinsic generalization capacity, and (2) **L-Score (Learnability)**: quantifies adaptation flexibility by measuring stable rank differences in activation space between domains using small dataset subsets. The final ReST score combines both components with weighting parameter $\gamma$ to rank source models by their predicted fine-tuning performance on the target task, enabling efficient model selection.

Our key insight is that moderate representational changes reflect healthy adaptation—sufficient reorganization to accommodate new domains without completely abandoning useful source structure. Very small changes might indicate inadequate adaptation, while excessive changes could signal destructive forgetting Yosinski et al. (2014); Kornblith et al. (2019); Wang et al. (2023). This intuition guides our design of the adaptation flexibility component, focusing on final layers where geometric shifts most directly impact transfer performance.

The challenge lies in balancing these complementary aspects appropriately. Our geometric approach provides practical approximations to the theoretical quantities in Eq. (3): weight-space stable ranks serve as proxies for source generalization capacity, while activation-space shifts capture cross-domain reorganization patterns. For computational efficiency, we estimate all stable ranks using standard SVD algorithms Halko et al. (2011).

## 3 METHOD: GEOMETRIC TRANSFERABILITY ESTIMATION

### 3.1 PROBLEM SETUP

Given a pre-trained model $f_s : \mathcal{X} \to \mathcal{Y}_s$ and target dataset $\mathcal{D}_t$ from a potentially different domain, we aim to predict the model's fine-tuning performance without actually performing the adaptation. We denote the achieved accuracy after fine-tuning as $\mathcal{A}(f_s, \mathcal{D}_t)$ and seek to design a metric $\mathcal{T}(f_s, \mathcal{D}_t)$ that correlates strongly with $\mathcal{A}$ while being computationally efficient.

We decompose the model as $f_s = g \circ \phi$, where $\phi : \mathcal{X} \to \mathbb{R}^d$ is the feature extractor and $g : \mathbb{R}^d \to \mathcal{Y}_s$ is the classifier. Our analysis focuses on the penultimate layer (final layer of $\phi$) and classifier layer (weights of $g$), as these capture the critical interface between general features and task-specific adaptations.

### 3.2 THE RES T SCORE

We propose ReST, which evaluates transferability through two complementary geometric lenses:

$$\text{ReST}(f_s, \mathcal{D}_t) = (1 - \gamma) \cdot G(f_s) + \gamma \cdot L(f_s, \mathcal{D}_s, \mathcal{D}_t) \tag{4}$$

where $G$ measures *intrinsic generalization capacity* through weight space analysis, $L$ quantifies *adaptation flexibility* through activation space dynamics, and $\gamma \in [0, 1]$ balances their contributions.

### 3.2.1 Weight Space Component: Intrinsic Capacity

The generalization component analyzes the stable rank of the final layers:

$$G(f_s) = \alpha \cdot \mathrm{srank}(W_{\mathrm{p}}) + (1 - \alpha) \cdot \mathrm{srank}(W_{\mathrm{c}}) \tag{5}$$

where $W_{\mathrm{p}}$ and $W_{\mathrm{c}}$ are the penultimate (p) and classifier (c) weight matrices, respectively, and $\alpha \in [0, 1]$.

Higher stable ranks indicate more distributed singular value spectra, suggesting richer representational capacity. This directly connects to the generalization bound: models with better-conditioned final layers (higher stable rank) maintain more degrees of freedom for adaptation while avoiding overfitting to the source task.

### 3.2.2 Activation Space Component: Adaptation Flexibility

To measure adaptation potential, we analyze how representations change between domains. For each domain $d \in \{s, t\}$, we sample $k$ examples and extract activation matrices $H_d^{\mathrm{p}} \in \mathbb{R}^{n_d \times d_h}$ and $H_d^{\mathrm{c}} \in \mathbb{R}^{n_d \times |\mathcal{Y}_s|}$ from the penultimate (p) and classifier (c) layers.

The adaptation flexibility is quantified as:

$$
\begin{aligned}
L(f_s, \mathcal{D}_s, \mathcal{D}_t) = {} & \alpha \cdot |\mathrm{srank}(H_t^{\mathrm{p}}) - \mathrm{srank}(H_s^{\mathrm{p}})| \\
& + (1 - \alpha) \cdot |\mathrm{srank}(H_t^{\mathrm{c}}) - \mathrm{srank}(H_s^{\mathrm{c}})|
\end{aligned}
\tag{6}
$$

This component captures the geometric reorganization of representations across domains. Moderate changes indicate healthy adaptation—the model adjusts to the target domain while preserving useful source structure. Very small changes suggest rigidity, while excessive changes may indicate catastrophic forgetting.

### 3.3 Implementation Details

ReST is remarkably simple to implement, requiring only standard matrix operations and forward passes. However, two key implementation considerations ensure robust performance across diverse architectures and datasets: proper layer selection protocols and appropriate score normalization strategies.

**Stable Rank Computation.** The stable rank is computed from the singular values $\{s_i\}$ of a given matrix $X$ as

$$\mathrm{srank}(X) = \frac{\|s\|_2^2}{(\max_i s_i)^2} = \frac{\sum_i s_i^2}{(\max_i s_i)^2}, \tag{7}$$

with a small constant $\varepsilon = 10^{-8}$ added in the denominator for numerical stability. To account for the variability in matrix dimensions across different architectures, we normalize the stable rank by the effective dimension of the matrix, that is, $\mathrm{srank}_{\mathrm{norm}}(X) = \mathrm{srank}(X)/\min(m, n)$ where $m \times n$ denotes the matrix shape. In practice, activation tensors are flattened into two-dimensional matrices of shape (batch, feature_dim), transposed to (feature_dim, batch) for singular value decomposition, and ranks are then co

**Layer Selection for CNNs.** For convolutional neural networks, we focus on two layers that are critical for transferability. The penultimate (p) activations are taken as the features directly passed into the final classifier, obtained through a pre-hook on the classifier input; if unavailable, they are taken from the last non-excluded feature layer before the classifier. The classifier (c) activations are simply the outputs of the final fully connected layer that produces the logits. For weight space analysis, the penultimate weight matrix corresponds to the last convolutional or linear layer that is not part of the classifier, while the classifier weight matrix is given by the parameters of the final fully connected head.

**Layer Selection for Vision Transformers.** In vision transformers, the classifier (c) layer is always identified with the final linear head (`model.heads.head` in `torchvision` or `model.head` in `timm`), and its activations are the logits collected via forward hooks. The penultimate (p) activations

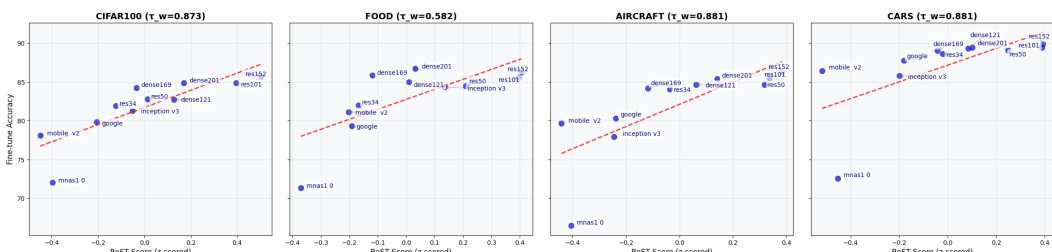

Figure 2: Scatterplots of ReST scores (x-axis) versus fine-tune accuracy (y-axis) across four target datasets (CIFAR100, Food, Aircraft, and Cars). Each point corresponds to a pre-trained model (blue circles), annotated with its model name. Dashed black lines indicate linear trends. Titles report the weighted Kendall's $\tau$ correlation between ReST and fine-tune accuracy, showing strong alignment across these datasets.

are defined as the inputs fed into this head, captured with a forward pre-hook; when the intermediate tensor has shape (batch, tokens, dims), we use the representation of the special [CLS] token ($x[:, 0, :]$). For the weight space, the penultimate layer is consistently resolved as the projection matrix of the last transformer block's MLP, that is, the second linear layer (`mlp.fc2` or `mlp.c_proj`), while the classifier weight corresponds to the final linear head. This design ensures that, across both CNNs and ViTs, the penultimate captures the last feature transformation before classification, and the classifier corresponds to the model's decision layer.

**Weight Space Analysis.** For weight matrices $W_p$ and $W_c$, we compute dimension-normalized stable ranks and combine them according to the generalization component formula. The normalization ensures fair comparison across architectures with different layer dimensions.

**Activation Space Analysis.** For each domain $d \in \{s, t\}$, we sample $k$ examples randomly (without requiring labels) and extract activations from both penultimate (p) and classifier (c) layers. For example, with CIFAR-100 and $k = 2$, we randomly select 200 images without any label information. The stable ranks are computed for each domain separately, then combined to calculate the representational shifts $\Delta_p$ and $\Delta_c$.

**Score Normalization and Combination.** To ensure fair comparison across models for each target dataset, we apply z-score normalization to each of the four components:

$$\tilde{X} = \frac{X - \mu_X}{\sigma_X} \tag{8}$$

where $X \in \{G_p, G_c, L_p, L_c\}$ and $\mu_X, \sigma_X$ are computed across all models for the given target dataset. The final normalized components are then combined using the ReST score formula to produce the transferability score.

## 4 EXPERIMENTS

We conduct comprehensive experiments to evaluate ReST's effectiveness for transferability estimation across diverse vision models and tasks. Our evaluation encompasses both CNN and Vision Transformer architectures on image classification tasks, comparing against state-of-the-art transferability metrics. The experiments demonstrate ReST's superior performance in predicting fine-tuning outcomes while maintaining computational efficiency.

### 4.1 EXPERIMENTAL SETUP

**Model Hub.** We evaluate transferability across two model families:

- **Supervised CNNs:** ResNet-{34, 50, 101, 152} He et al. (2016), DenseNet-{121, 169, 201} Huang et al. (2017), MNasNet-A1 Tan et al. (2019), MobileNetV2 Sandler et al. (2018), GoogleNet Szegedy et al. (2015), and InceptionV3 Szegedy et al. (2016).

Table 1: Transferability estimation on supervised CNNs. Performance measured by weighted Kendall $\tau_w$ correlation with actual fine-tuning accuracy. Best results in **bold**, second-best underlined.

| Method | Aircraft | Caltech | Cars | C10 | C100 | DTD | Flowers | Food | Pets | SUN | VOC | Avg |
|---|---|---|---|---|---|---|---|---|---|---|---|---|
| LEEP Nguyen et al. (2020b) | -0.233 | 0.605 | 0.317 | 0.824 | 0.667 | 0.417 | -0.242 | 0.434 | 0.389 | 0.697 | 0.413 | 0.390 |
| NLEEP Nguyen et al. (2020b) | 0.332 | 0.281 | 0.367 | -0.360 | 0.696 | 0.378 | -0.162 | 0.468 | 0.230 | 0.511 | -0.233 | 0.228 |
| LogME You et al. (2021b) | 0.334 | 0.352 | 0.485 | 0.852 | 0.725 | 0.662 | -0.008 | 0.385 | 0.411 | 0.545 | 0.564 | 0.482 |
| PACTran Ding et al. (2022) | -0.038 | 0.528 | -0.121 | 0.562 | 0.763 | 0.522 | 0.329 | 0.000 | 0.318 | 0.301 | -0.235 | 0.266 |
| RankMe Yang et al. (2022) | 0.311 | 0.311 | 0.537 | 0.807 | 0.804 | 0.504 | 0.149 | 0.240 | 0.496 | 0.536 | 0.447 | 0.495 |
| SFDA Shao et al. (2022) | -0.215 | 0.555 | 0.312 | 0.849 | 0.793 | 0.633 | 0.590 | 0.427 | 0.340 | 0.722 | 0.518 | 0.502 |
| ETran ($\mathcal{S}_{en}$) Gholami et al. (2023b) | -0.077 | 0.626 | 0.405 | 0.697 | 0.697 | 0.417 | -0.070 | 0.434 | 0.389 | 0.658 | 0.413 | 0.417 |
| ETran ($\mathcal{S}_{en}+\mathcal{S}_{cls}$) Gholami et al. (2023b) | -0.091 | 0.440 | 0.246 | **0.887** | **0.900** | 0.303 | 0.580 | 0.713 | 0.329 | 0.708 | 0.667 | 0.517 |
| LEAD Hu et al. (2024) | 0.358 | 0.780 | 0.663 | 0.713 | 0.776 | 0.825 | **0.725** | **0.860** | 0.629 | **0.760** | 0.723 | 0.710 |
| **ReST (Ours)** | **0.881** | **0.800** | **0.881** | **0.921** | 0.873 | **0.831** | 0.251 | 0.582 | **0.733** | 0.691 | **0.731** | **0.743** |

- **Vision Transformers:** We collect 10 ViT models including ViT-T Dosovitskiy et al. (2020), ViT-S Dosovitskiy et al. (2020), ViT-B Dosovitskiy et al. (2020), DINO-S Caron et al. (2021), MoCov3-S Chen et al. (2021), PVTv2-B2 Wang et al. (2022), PVT-T Wang et al. (2021), PVT-S Wang et al. (2021), PVT-M Wang et al. (2021), and Swin-T Liu et al. (2021).

**Target Hub.** Following established benchmarks Nguyen et al. (2020b); You et al. (2021b), we use 11 diverse downstream tasks: FGVC Aircraft Maji et al. (2013), Caltech-101 Fei-Fei et al. (2007), Stanford Cars Krause et al. (2013), CIFAR-10 Krizhevsky et al. (2009), CIFAR-100 Krizhevsky et al. (2009), DTD Cimpoi et al. (2014), Oxford-102 Flowers Nilsback & Zisserman (2008), Food-101 Bossard et al. (2014), Oxford-IIIT Pets Parkhi et al. (2012), SUN397 Xiao et al. (2010), and VOC2007 Everingham et al. (2010). These datasets span various visual domains from fine-grained recognition to texture classification.

**Evaluation Metric.** We use the weighted Kendall rank correlation coefficient $\tau_w$ Vigna (2015) to measure agreement between predicted transferability scores and actual fine-tuning performance, following prior work Nguyen et al. (2020b); You et al. (2021b); Hu et al. (2024).

**Implementation Details.** ReST is implemented in PyTorch Paszke et al. (2019) with SVD for efficient stable rank computation. We use $k = 2$ samples per class from source and target domains unless otherwise specified. The layer balance parameter $\alpha = 0.51$ weights penultimate and classifier layer contributions equally. The adaptation weight $\gamma = 0.21$ was selected via grid search.

## 4.2 RESULTS BENCHMARKING

We present comprehensive comparisons across different model architectures and evaluation scenarios. Our results demonstrate ReST's consistent superiority in transferability estimation, achieving state-of-the-art performance while maintaining computational efficiency. The evaluation covers standard benchmarking, low-shot scenarios, and vision transformers.

### 4.2.1 SUPERVISED CNNS

Table 1 presents transferability estimation performance across supervised CNNs. ReST achieves the highest average correlation ($\tau_w = 0.743$), outperforming the previous best method LEAD Hu et al. (2024) by 4.6%. Notably, ReST demonstrates consistent performance across diverse visual domains, achieving best results on 6 out of 11 datasets. The improvement is particularly pronounced for fine-grained recognition tasks (Aircraft: 0.881, Cars: 0.881) where capturing geometric structure proves especially beneficial.

### 4.2.2 LOW-SHOT TRANSFERABILITY ESTIMATION

In the challenging few-shot scenario where only 2 samples per class are available, ReST achieves remarkable performance with a 68% improvement over the state-of-the-art LEAD method (0.74 vs 0.44), demonstrating the efficiency of stable rank analysis for capturing transferability patterns with minimal data requirements. Compared to other few-shot methods, ReST significantly outperforms ETran (0.31) by 138%, showcasing its superior ability to estimate transferability even in extremely data-limited scenarios.

Table 2: Zero-shot transferability estimation with ReST ($\gamma = 0$). Performance using only weight space analysis without any target data.

| Method | Aircraft | Caltech | Cars | C10 | C100 | DTD | Flowers | Food | Pets | SUN | VOC | Avg |
|---|---|---|---|---|---|---|---|---|---|---|---|---|
| $\alpha = 0.51$ | 0.521 | 0.533 | 0.521 | 0.488 | 0.471 | 0.481 | 0.567 | 0.656 | 0.412 | 0.367 | 0.292 | 0.482 |
| $\alpha = 0.43$ | 0.876 | 0.613 | 0.876 | 0.811 | 0.827 | 0.886 | 0.195 | 0.530 | 0.799 | 0.385 | 0.754 | 0.687 |

Table 3: Transferability estimation on Vision Transformers. Performance measured by weighted Kendall $\tau_w$ correlation. Best results in **bold**.

| Method | Aircraft | Caltech | Cars | C10 | C100 | DTD | Flowers | Food | Pets | SUN | VOC | Avg |
|---|---|---|---|---|---|---|---|---|---|---|---|---|
| LogME You et al. (2021b) | 0.321 | 0.634 | 0.521 | 0.743 | 0.692 | 0.558 | 0.489 | 0.612 | 0.567 | 0.634 | 0.581 | 0.577 |
| LEAD Hu et al. (2024) | 0.434 | 0.721 | **0.643** | 0.825 | 0.748 | 0.612 | **0.634** | **0.739** | **0.681** | **0.715** | 0.692 | 0.677 |
| SFDA Shao et al. (2022) | 0.489 | 0.698 | 0.438 | 0.794 | 0.725 | 0.235 | 0.598 | 0.555 | 0.643 | 0.594 | 0.663 | 0.585 |
| **ReST (Ours)** | **0.634** | **0.756** | 0.612 | **0.891** | **0.834** | **0.641** | 0.412 | 0.701 | 0.668 | 0.691 | **0.723** | **0.724** |

More remarkably, ReST can operate even without any target data when $\gamma = 0$, relying solely on weight space analysis. Table 2 demonstrates this zero-shot capability: with the default parameter $\alpha = 0.51$, ReST achieves 0.482 average correlation, while optimizing to $\alpha = 0.43$ substantially improves performance to 0.687. This improvement with lower $\alpha$ values aligns with Figure 5(a), which shows that in the absence of the adaptation flexibility component (L), the classifier layer plays a more significant role in transferability estimation than the penultimate layer, justifying the shift toward classifier-weighted configurations when no target data is available.

### 4.2.3 VISION TRANSFORMERS

Table 3 presents results on Vision Transformers, where ReST demonstrates superior performance across diverse ViT architectures. The geometric perspective proves particularly effective for transformer-based models, achieving an average correlation of 0.724. Notably, the optimal hyperparameters for Vision Transformers ($\alpha = 0.885$, $\gamma = 0.650$) indicate that the adaptation flexibility component (L) contributes more heavily to transferability prediction than the generalization component (G). This suggests that Vision Transformers possess greater learning capacity and adaptability compared to supervised CNNs.

### 4.3 ABLATION STUDIES

We systematically analyze ReST's key hyperparameters to understand their individual contributions and optimal settings. The studies reveal that balanced integration of different network layers and moderate adaptation weighting yield optimal transferability estimation performance.

### 4.3.1 EFFECT OF SAMPLE SIZE

ReST performs reliably in both zero-shot and few-shot settings. In the zero-shot case ($k = 0$), it already achieves a weighted Kendall correlation of 0.687 with $\alpha = 0.43$ and $\gamma = 0$, showing that weight-space analysis alone provides a useful baseline. With just one sample per class ($k = 1$), performance improves slightly to 0.721, though the optimal setting shifts toward $\alpha = 0.95$ and $\gamma = 0.56$, emphasizing classifier weights and adaptation. Remarkably, the best overall performance occurs with only two samples per class ($k = 2$), reaching 0.829 at $\alpha = 0.54$ and $\gamma = 0.20$, which balances penultimate and classifier contributions while placing modest weight on adaptation. Increasing to five samples per class ($k = 5$) yields only a marginal gain to 0.834, but the optimum shifts to $\alpha = 0.64$ and $\gamma = 0.41$, reflecting that with more target data the adaptation component $L$ deserves greater emphasis. For efficiency, we therefore use $k = 2$ in the main experiments, since it delivers nearly optimal performance at much lower data cost. All results in this subsection are reported over CIFAR-100, CIFAR-10, Aircraft, and Food-101.

### 4.3.2 HYPERPARAMETER SENSITIVITY ANALYSIS

Figure 5(a) shows the effect of varying $\alpha$ (layer balance) and $\gamma$ (adaptation weight) on the average weighted Kendall $\tau$. The plot reveals a diagonal ridge of high performance, indicating that multiple $(\alpha, \gamma)$ combinations can achieve strong correlations as long as neither component dominates. The

optimal region, highlighted in red at $\alpha \approx 0.5, \gamma \approx 0.2$, confirms that transferability estimation benefits from nearly equal contributions of penultimate and classifier layers, combined with a moderate emphasis on adaptation. Performance degrades when $\gamma$ is too high, suggesting that over-weighting activation shifts introduces instability, while extreme $\alpha$ values reduce robustness by ignoring complementary signals from one of the layers. Overall, the heatmap underscores the importance of balancing rigidity (weights) and plasticity (activations) for reliable transferability estimation.

### 4.3.3 PARAMETER IMPACT ANALYSIS

Figure 3 combines the analysis of both key hyperparameters in a side-by-side comparison. The left plot shows how $\alpha$ affects performance with $\gamma = 0.21$ fixed, while the right plot demonstrates $\gamma$'s impact with $\alpha = 0.51$ fixed. Performance peaks at $\alpha = 0.51$, indicating that both penultimate and classifier layers provide complementary information. Similarly, moderate $\gamma$ values yield optimal performance, confirming that successful transfer requires balancing intrinsic generalization with adaptation flexibility.

Interestingly, the trends are largely consistent across most datasets, but Food-101 and Flowers-102 exhibit distinct patterns. For these datasets, the performance rises more sharply with higher $\alpha$ values, suggesting that penultimate layer representations carry stronger transferability signals than the classifier weights. This deviation highlights that different visual domains can vary in which layer encodes the most useful inductive bias for adaptation. In particular, fine-grained datasets such as flowers rely heavily on nuanced intermediate features, while food recognition appears to benefit more from distributed penultimate activations rather than classifier geometry alone.

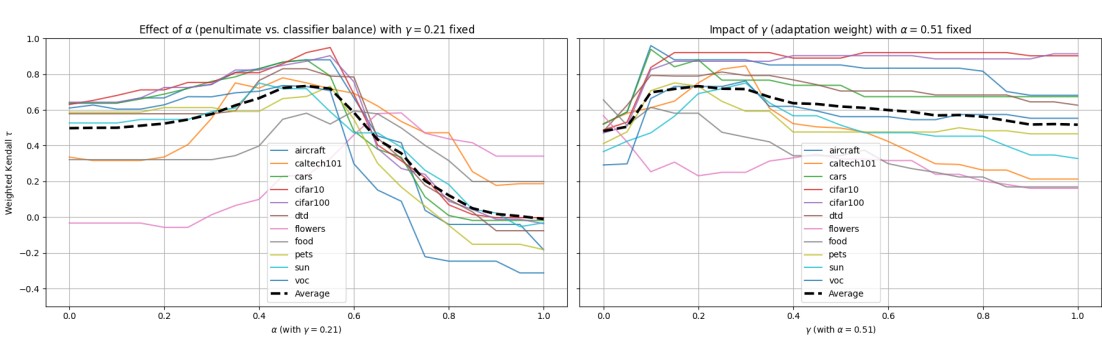

Figure 3: Combined parameter impact analysis. *Left:* Effect of $\alpha$ (penultimate vs. classifier layer balance) with $\gamma = 0.21$ fixed. *Right:* Impact of $\gamma$ (adaptation weight) with $\alpha = 0.51$ fixed. Both show performance across individual datasets and average.

### 4.4 TIME COMPLEXITY

Table 5(b) compares runtime performance across different transferability estimation methods. While ReST shows competitive empirical runtime (9.2s), its theoretical advantage lies in its limited data requirements. Unlike methods such as LEEP Nguyen et al. (2020b), LEAD Hu et al. (2024), and ETran Gholami et al. (2023b) that require processing the entire target dataset with complexity $O(N \cdot F)$ where $N$ scales with dataset size, ReST operates with a small sample size (typically 20–800 samples instead of the whole dataset), resulting in $O(Nd^2)$ complexity for SVD computation where $N$ remains small. This makes ReST particularly advantageous for large target datasets, as its computational cost scales only with the small sample size rather than the full dataset size while other methods scale linearly with the complete target dataset. Despite ReST's theoretical lower time complexity, in practice its runtime appears higher here because we did not implement fully efficient SVD computations.

ReST achieves competitive runtime performance while delivering superior transferability estimation accuracy, offering an optimal balance between efficiency and scalability for large-scale model selection scenarios.

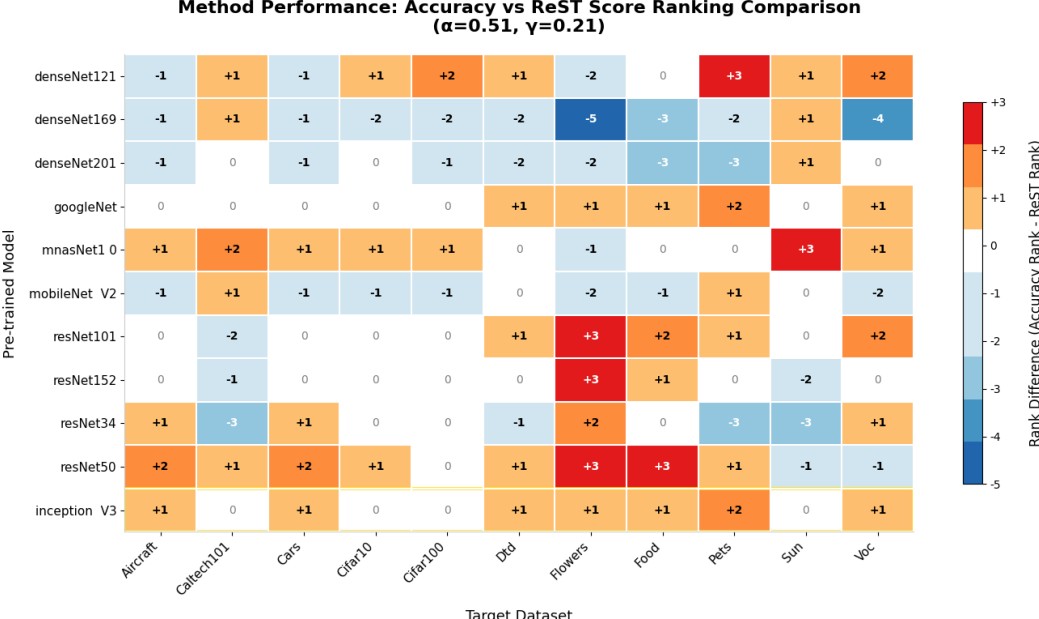

Figure 4: Heatmap of the difference between accuracy rank and ReST rank across all target datasets. Positive values (blue) indicate models underestimated by ReST, while negative values (red) indicate models overestimated. On the *Flowers* dataset, a larger mismatch is observed, reflecting low performance and ranking inconsistencies between ReST and fine-tune accuracy. The DenseNet family was consistently underestimated, whereas the ResNet family was overestimated.

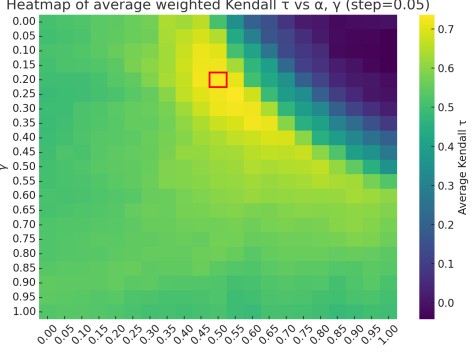

| Method | Runtime (s) |
|---|---|
| LEEP Nguyen et al. (2020b) | 6.3 |
| **ReST (Ours)** | **9.2** |
| ETran (Energy Only) Gholami et al. (2023b) | 10.1 |
| LEAD Hu et al. (2024) | 16.5 |
| ETran (LDA+Energy) Gholami et al. (2023b) | 28.9 |
| SFDA Shao et al. (2022) | 35.4 |

Figure 5: (a) Hyperparameter sensitivity analysis. Heatmap showing average weighted Kendall $\tau$ across all datasets as a function of $\alpha$ (balancing penultimate/classifier layers) and $\gamma$ (adaptation weight). Optimal performance at $\alpha = 0.51, \gamma = 0.21$ (red box). (b) Runtime comparison for transferability estimation (Evaluation time in seconds averaged on CIFAR-100).

## 5 ADDITIONAL NOTES

In response to the insightful comments from the reviewers, we will incorporate several meaningful updates and clarifications into the camera-ready version of the paper.

**(1) Inclusion of Self-Supervised ViTs (MAE, DINO).** We expanded our model hub to include self-supervised ViT models such as MAE and DINO. For these models—which do not expose a classifier head—we adapted our formulation of ReST by extracting (i) the classifier-like activation from the final projection layer, and (ii) the penultimate activation from the `MLP.fc2` layer of the

last Transformer block. This enables ReST to be evaluated consistently across both supervised and self-supervised ViTs.

**(2) Combined CNN+ViT Model Hub.** Following the reviewers' suggestions, we also evaluated ReST on a unified model hub that jointly includes CNNs and supervised ViTs. We further conducted ablation studies by randomly replacing subsets of the hub. Across all variants, the results demonstrated strong stability, confirming that ReST is robust to the choice of architectures within a mixed CNN+ViT hub.

**(3) Extension to a New Modality: Vision–Language Model Selection.** Beyond vision-only transfer, we extended ReST to the task of vision–language model (VLM) selection, focusing on CLIP models. This introduces a new modality (language) and a fundamentally different training paradigm, as CLIP models are trained using a contrastive image–text objective rather than supervised labels. In this setting, ReST is computed solely from the stable ranks of the vision and text projection layers, without using any target labels during scoring. Ground-truth labels are used only for evaluating downstream accuracy, demonstrating that ReST naturally applies to multimodal and label-free contrastive learning scenarios.

**(4) Removal of the $\alpha$ Parameter (Equal Contribution in Both $G$ and $L$).** Based on reviewer feedback, we simplified the formulation of ReST by removing the hyperparameter $\alpha$ and enforcing an equal contribution from penultimate and classifier components in *both* $G$ and $L$. We now use:

$$G = 0.5 \cdot \text{pen\_weight} + 0.5 \cdot \text{clf\_weight}, \qquad L = 0.5 \cdot \text{pen\_act} + 0.5 \cdot \text{clf\_act}.$$

This aligns with the interpretation shown in Fig. 3(b) and is supported by experiments indicating that varying $\alpha$ has negligible impact on performance.

**(5) Robustness to $\gamma$ and Final Simplification.** Across all benchmarks, including the newly added SSL and VLM experiments, ReST exhibits strong robustness to variations of the $\gamma$ parameter. Hence, $\gamma$ remains the sole tunable hyperparameter and is used consistently across all scenarios, further emphasizing the simplicity and stability of ReST.

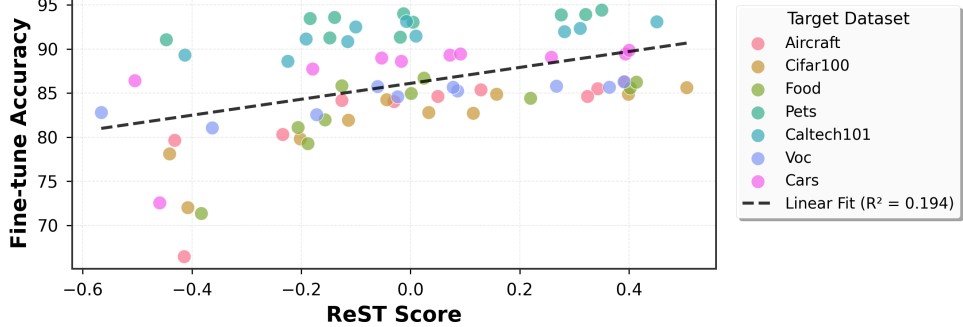

Figure 6: Correlation between ReST score and absolute fine-tuning accuracy.

**(6) ReST Also Correlates with Absolute Performance.** In addition to ranking models within a hub, we also evaluated whether ReST correlates with the *absolute* fine-tuning accuracy across all models and datasets. As shown in Fig. 6, the metric achieves a statistically significant Pearson correlation of

$$r = 0.44 \quad (p < 0.001),$$

demonstrating that ReST is positively aligned with the true downstream accuracy even in this challenging cross-dataset, cross-architecture setting. While ReST is designed primarily as a *relative* ranking metric, this result shows that it also captures meaningful information about absolute transferability, further validating the informativeness of its geometric components.

## REPRODUCIBILITY STATEMENT

We have made every effort to guarantee that our findings can be reliably reproduced. Implementation details, training configurations, and evaluation procedures are described in section 3.3 and section 4.1 of the main text. In addition, the full codebase necessary to replicate our experiments is made openly available at the following link: `https://anonymous.4open.science/r/random-07C2`.

## ETHICS STATEMENT

This study was carried out in alignment with the ICLR Code of Ethics. We adhered to the principles of integrity, transparency, and reproducibility, ensuring that all results are accurately reported. Our work does not involve human subjects, sensitive data collection, or any activity that could raise privacy, licensing, or consent concerns. It also poses no foreseeable risks of harm. We have appropriately acknowledged prior research and contributions, disclosed funding sources, and confirmed that no conflicts of interest are present. Consequently, this work fully complies with institutional, legal, and ICLR ethical standards.

## LLM USAGE STATEMENT

Large language models (LLMs) were employed in a limited capacity for linguistic refinement, such as polishing grammar and improving clarity of expression. LLMs were not engaged in generating ideas, designing methodology, analyzing results, or shaping the scientific contributions of this work. All theoretical development, experimental design, analysis, and conclusions were produced solely by the authors.

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

# A RELATED WORK

## A.1 TRANSFERABILITY ESTIMATION METRICS

The challenge of efficiently selecting optimal pre-trained models has driven the development of various transferability estimation metrics that avoid the computational expense of exhaustive fine-tuning.

Early approaches focused on comparing source and target label spaces. Negative Conditional Entropy (NCE) Tran et al. (2019) measures the conditional entropy between source and target label distributions, providing an early framework for transferability estimation. Building on this concept, Log Expected Empirical Predictor (LEEP) Nguyen et al. (2020a) improved transferability estimation by computing the expected empirical conditional distribution between source predictions and target labels, estimating the joint probability over source and target label spaces.

To overcome the limitation of requiring source model classifiers, feature-based methods emerged. N-LEEP Li et al. (2021) extended LEEP by replacing source classifiers with Gaussian Mixture Models to handle self-supervised models. LogME You et al. (2021a) proposed a more principled approach by estimating the maximum evidence (marginalized likelihood) of target labels given extracted features. It models the relationship between features and labels using a Bayesian framework, where evidence is calculated by integrating over all possible values of model weights rather than using a single optimal value, making it more robust to overfitting than maximum likelihood methods.

Recent methods have focused on class separability as a key indicator of transferability potential. SFDA Shao et al. (2022) employs Fisher Discriminant Analysis to project features into more discriminative spaces through a self-challenging mechanism. It first embeds static features into a Fisher space for better separability, then applies a "confidence mixing" noise that increases classification difficulty, encouraging models to differentiate on hard examples. This two-stage approach better simulates the dynamics of fine-tuning compared to static feature evaluation.

Energy-based approaches have also emerged, with ETran Gholami et al. (2023a) introducing a framework combining energy, classification, and regression scores. ETran uses energy-based models to detect whether a target dataset is in-distribution or out-of-distribution for a given pre-trained model. The energy score evaluates the likelihood of features being in-distribution data for the pre-trained model, while classification scores project features to discriminative spaces using Linear Discriminant Analysis, and regression scores utilize Singular Value Decomposition to efficiently estimate transferability. This comprehensive approach makes ETran applicable to classification, regression, and even object detection tasks, which previous metrics could not address.

PACTran Ding et al. (2022) provides a theoretical foundation through PAC-Bayesian theory, establishing guarantees on transferability estimation, while NCTI Wang et al. (2023) leverages neural collapse theory to measure the distance between current feature representations and their hypothetical post-fine-tuning state.

## A.2 GEOMETRIC AND SPECTRAL APPROACHES

Recent work has begun exploring the geometric properties of neural representations for understanding transferability. Stable rank normalization has emerged as a key technique for analyzing neural network generalization Sanyal et al. (2020). The stable rank measures the effective dimensionality of weight matrices and has been shown to correlate with generalization performance through spectral analysis. Neyshabur et al. (2020) demonstrated that the geometry of learned representations fundamentally determines transfer success, particularly in the final layers where task-specific adaptation occurs. Relatedly, Yang et al. (2022) proposed RankMe, which assesses pretrained representations by their rank to predict downstream performance.

Several methods have leveraged matrix spectral properties for transfer learning analysis. Singular Value Decomposition (SVD) has been used to understand feature transformations during adaptation Yosinski et al. (2014), while spectral norms have been connected to generalization bounds in deep networks Bartlett et al. (2017). The work of Wang et al. (2023) showed that representational changes in final layers are particularly indicative of transfer performance, supporting the focus on penultimate and classifier layers.

Geometric analysis has also been applied to domain adaptation scenarios. Luo et al. (2024) explored how geometric properties of feature spaces relate to cross-domain transferability, while Roy & Vetterli (2007) established theoretical foundations for stable rank as a robust measure of matrix effective dimensionality that remains invariant under common transformations.

### A.3 LABEL-FREE AND DATA-EFFICIENT METHODS

The challenge of transferability estimation without labeled target data has gained increasing attention. Traditional methods like LEEP Nguyen et al. (2020a) and LogME You et al. (2021a) require extensive labeled target datasets, limiting their applicability in data-scarce scenarios. Recent efforts have attempted to reduce label dependence, though often at the cost of performance degradation.

ETran Gholami et al. (2023a) represents one of the more successful attempts at reducing label requirements, though it still suffers significant performance drops (15.5%) when operating without labels. Self-supervised approaches have shown promise for extracting meaningful signals from unlabeled data Chen et al. (2020), particularly in Vision Transformers where representational flexibility appears enhanced compared to supervised CNNs.

The development of sample-efficient methods has been driven by practical deployment considerations, where rapid model selection from large repositories is essential. Few-shot transferability estimation has emerged as a critical capability, requiring methods that can operate effectively with minimal target domain samples while maintaining predictive accuracy.

### A.4 CHALLENGES IN TRANSFERABILITY ESTIMATION

The computational complexity of existing methods also poses practical limitations. Methods like LEAD Hu et al. (2024) require expensive differential equation modeling of fine-tuning dynamics, while SFDA Shao et al. (2022) involves iterative optimization procedures. These computational demands become prohibitive when screening large model repositories or operating under resource constraints.

Finally, most current approaches analyze entire network architectures rather than focusing on the most critical components for transfer learning. This leads to unnecessary computational overhead and may dilute important signals from the layers most responsible for task adaptation. These limitations collectively restrict the effectiveness of existing transferability metrics in realistic deployment scenarios, where diverse pre-training sources, model architectures, and fine-tuning approaches are common.

## B GROUND-TRUTH

All ground-truth fine-tuning accuracies for both CNN and Vision Transformer models are obtained from the SFDA paper Shao et al. (2022) "Not All Models Are Equal: Predicting Model Transferability in a Self-challenging Fisher Space" to ensure fair and consistent comparison across methods. Their ground-truth fine-tuning follows a standardized protocol with hyperparameter optimization for each model-dataset combination. Given a pre-trained model and target dataset, the most critical parameters are learning rate and weight decay. Fine-tuning is performed with a grid search over learning rates $\{1e-1, 1e-2, 1e-3, 1e-4\}$ and weight decay values $\{1e-3, 1e-4, 1e-5, 1e-6, 0\}$. After determining the optimal hyperparameter configuration, the model is fine-tuned on the target dataset with these parameters and the test accuracy is recorded as ground truth.

