# OpenReview forum: "ReST: Remarkably Simple Transferability Estimation"
_ICLR.cc/2026/Conference — Submitted to ICLR 2026_

### Official Review · Reviewer_39ug · 2025-10-26

**Soundness:** 2
**Presentation:** 3
**Contribution:** 2
**Rating:** 4
**Confidence:** 3

**Summary:**

This paper introduces ReST (Remarkably Simple Transferability Estimation), a novel method designed to predict the fine-tuning performance of pre-trained neural networks on target tasks without requiring extensive experimentation or labeled target data. ReST is positioned as a simple yet effective approach that analyzes the geometric structure of representations, specifically using stable rank—a measure of matrix effective dimensionality. The ReST score combines two components: the Generalization (G-Score), based on the stable rank of the penultimate ($W_p$) and classifier ($W_c$) weight matrices to measure intrinsic capacity, and the Learnability (L-Score), which quantifies adaptation flexibility by measuring the relative change in stable rank of activation features between source and target domains. The method is computationally efficient, reducing to simple matrix norm calculations.

**Strengths:**

**Geometric Insight**: The approach is fundamentally insightful, linking transfer success to two geometric properties captured by stable rank: intrinsic generalization capacity (G) (distributed weight spectra) and adaptation flexibility (L) (moderate activation shifts). This provides valuable insight into why certain representations transfer well.

**Remarkable Data Efficiency**: ReST is a label-free estimator requiring only a small, unlabeled sample subset (effective size equivalent to $\approx 2$ samples per class). This addresses a key practical limitation of prior methods like LEEP and LogME, which require substantial labeled data.

**Weaknesses:**

**Hyperparameter Sensitivity and Complexity Trade-off**: The "Remarkably Simple" title is somewhat undermined by the reliance on architecture-specific tuning of the layer balance parameter ($\alpha$) and adaptation weight ($\gamma$) for optimal performance. For instance, optimal parameters shift significantly between supervised CNNs ($\alpha=0.51, \gamma=0.21$) and self-supervised Vision Transformers ($\alpha=0.885, \gamma=0.650$). Furthermore, zero-shot performance requires changing $\alpha$ from 0.51 to 0.43 to achieve the highest correlation (0.687). This suggests that a user needs to pre-determine the model class or data regime to select the optimal configuration, adding a layer of hidden complexity to the "simple" methodology. It’s also unclear how these values are chosen, if these values were chosen based on correlation with downstream test accuracy, this is unjustified. Please shed light on the exact procedure for selecting these hyperparameters in the particular problem setting.

**Ranking Inconsistencies**: The heatmap analysis (Figure 4) explicitly reveals that ReST shows a larger mismatch (ranking inconsistency) on specific datasets like Flowers-102, and consistently underestimates the DenseNet family while overestimating the ResNet family. While overall performance is SOTA, these systematic biases warrant deeper investigation.

**Relevant Literature missing**: Some relevant work on transferability measures is missing from related work. E.g., H-Score [Bao 2019, Ibrahim 2022]. Please include these in the related work. If possible, include these in the comparisons as well.

**Typos and Formatting Issues**
- **Source Labeling/Numbering**: The text references "Table 5(b)", but the corresponding comparison table is labeled "Figure 5: (b) Runtime comparison...". This confusion between table and figure numbering should be corrected for consistency (e.g., Table 5).
- **Reference to Figure 5(a)**: In Section 4.3.2, the text discusses "Figure 5(a) shows the effect..." but Figure 5 is located much later. This placement or numbering should be improved.

*References*:
- Yajie Bao et al. “An Information-Theoretic Approach to Transferability in Task Transfer Learning.” 2019 IEEE International Conference on Image Processing (ICIP) (2019): 2309-2313.
- Shibal Ibrahim, Natalia Ponomareva, and Rahul Mazumder. 2022. Newer is Not Always Better: Rethinking Transferability Metrics, Their Peculiarities, Stability and Performance. In Machine Learning and Knowledge Discovery in Databases: European Conference, ECML PKDD 2022.

**Questions:**

**Hyperparameter Generalization**: The optimal parameters ($\alpha, \gamma$) vary significantly based on the model type (CNN vs. ViT) and data availability (zero-shot vs. few-shot). Can the authors propose a robust, default setting for $\alpha$ and $\gamma$ that performs consistently well across all architectures and regimes (perhaps slightly below the peak optima) to truly deliver a "Remarkably Simple" out-of-the-box solution without model-specific tuning?

**Normalization of L-Score**: The normalization of scores across the models seems very hacky to me. Isn’t there a better way of normalization e.g., for $L$ relative change divided by source stable rank, especially given that activation stable rank values might vary widely depending on the model's depth or architecture? Or some other alternatives?

---

> ### Author Response · Authors · 2025-11-25
>
> We thank the reviewer for the valuable feedback. We provided some experiments and explanations in the other comments, and here we address your concerns:
>
> **Hyperparameter Generalization**
>
> *Equal Contribution of Penultimate and Classifier Components.*
> Based on the reviewer's feedback, we simplified the formulation of ReST by removing the hyperparameter $\alpha$ and enforcing an equal contribution from the penultimate and classifier components in *both* $G$ and $L$.
>
> Specifically, we now use the fixed formulation:
>
> $$G = 0.5 \cdot \text{pen-weight} + 0.5 \cdot \text{clf-weight}$$
>
> $$L = 0.5 \cdot \text{pen-act} + 0.5 \cdot \text{clf-act}.$$
>
> This choice matches the interpretation used in Fig. 3(b), where equal weighting across components yields the most stable results across all tasks.
>
> *Final Simplification and Retention of a Single Hyperparameter.*
> We removed $\alpha$ entirely from the method; after this simplification, $\gamma$ remains the *only* tunable hyperparameter in ReST, controlling the balance between the weight-space ($G$) and activation-space ($L$) components.
> As shown in Fig. 3(b), ReST exhibits strong robustness to variations of $\gamma$—both in terms of the average performance and consistently across all individual datasets.
>
>
> In the classical transferability-estimation setup, it is typically assumed that we have access to the target features, target labels, and a collection of pretrained models. Most prior methods (e.g., LEEP, LogME, LEAD, SFDA) explicitly rely on the full labeled target dataset to compute their scores. In contrast, ReST does not require any target labels: using only a very small number of unlabeled target samples, our simple geometric score consistently outperforms state-of-the-art methods that use far more information. We also explored an even more restrictive scenario where no target features are available at all. In this case, the activation-based component cannot be computed, and thus we naturally set $\gamma = 0$, relying solely on the weight-based term. Even under this extreme zero-feature setting, ReST maintains strong performance. Finally, in the revised version we fixed $\alpha = 0.5$, ensuring equal contribution from the penultimate and classifier layers. This means that practitioners can use a single, unified configuration of ReST across both settings—whether unlabeled target samples are available or not—making the method simple, robust, and easy to apply in practice.
>
> We thank the reviewer for the valuable feedback. We provided some experiments and explanations in the other comments, and here we address your concerns:
>
> When we applied your valued suggestion of the normalization method, the correlation between changes in ReST and changes in accuracy decreased to 0.08. This indicates that normalizing the final scores and changes appears to work better.
>
> **Absolute vs. Relative Performance Correlation for Normalized Score**
>
> In addition to ranking models within a hub, we also evaluated whether ReST correlates with the *absolute* fine-tuning accuracy across all models and datasets. The corresponding results are provided at line 529 of the revised paper. As shown in Fig. 6, ReST achieves a statistically significant Pearson correlation of
> $$r = 0.44 \quad (p < 0.001),$$
> demonstrating that the score not only preserves reliable *relative* rankings within each model hub but also aligns meaningfully with true downstream accuracy in an absolute sense. This global correlation confirms that the geometric signals used by ReST generalize across datasets, architectures, and training paradigms, further validating the informativeness of the metric.

---

### Official Review · Reviewer_ki9i · 2025-10-31

**Soundness:** 2
**Presentation:** 3
**Contribution:** 2
**Rating:** 2
**Confidence:** 4

**Summary:**

This paper proposes a transferability estimation (TE) method named ReST for evaluating the downstream task transferability of pre-trained models. The core idea of ReST is to analyze the generalization score (G-Score) and learnability score (L-Score) of the final two layers of the model, achieving SOTA results with a small computational cost using only a small amount of unlabeled data.

**Strengths:**

ReST is simple and does not use annotations.

**Weaknesses:**

-- ReST requires the use of a classifier layer, which tends to limit its wider application, such as in self-supervised models like MAE and DINO that do not have a classifier layer.

-- The method presented in this paper is relatively innovative. It mainly focuses on the combination and extension of existing methods (Stable rank).

-- The figures are blurry, the font size within the images is too small, and the layout of elements within the figures is unclear.

-- The ablation studies seem to show that ReST is not stable (i.e., it is quite sensitive to hyperparameters). Although Figure 5(a) illustrates the impact of the two hyperparameters, Table 2 shows that a small change in $\alpha$ can cause large fluctuations in results for specific datasets. For instance, when $\alpha=0.43$, ReST performs significantly better on the Aircraft and Cars datasets, but its performance drops markedly on the Flowers dataset.

-- Although most TE methods require using a large amount of downstream data to input and extract features while ReST only needs two unlabeled samples, other TE methods achieve plug-and-play based on features. In contrast, ReST needs to access the model's internal weights and activation values, which might be a disadvantage.

-- The paper's intuition for the L-Score appears to contradict its final formulation. In Section 3.2.2, the authors state that moderate representational changes indicate healthy adaptation, whilea very large L-Score could signal catastrophic forgetting. However, the final ReST score (Eq. 4) scales the L-Score with a simple positive coefficient ($\gamma$).

**Questions:**

-- Although the experimental setup of this paper is quite similar to previous methods, I believe that CNN and ViT should not be evaluated separately. How about combining the experiments for CNN and ViT? This is closer to the actual usage scenario. No additional experiments are required. It is sufficient to combine the experimental results and recalculate the Kendall coefficient.

-- Existing self-supervised ViT models have been widely applied. How can ReST be utilized in these models? Have you considered evaluating on these newer and better models?

---

> ### Author Response · Authors · 2025-11-25
>
> We thank the reviewer for the thoughtful and constructive feedback. We address each weakness and question below.
>
> *"ReST requires the use of a classifier layer... limits application to self-supervised models like MAE and DINO."*
>
> Thank you for pointing this out. We have added an evaluation scenario in which the model hub contains self-supervised models (MAE and DINO) as well, as suggested:
> ```
> mae_vitb16, mae_vitl16, dino_vits8, dino_vitb16
> ```
>
> **How ReST is computed for SSL models (no classifier head).**
> Self-supervised models do not include a classifier layer, but they do expose the components needed for our formulation of ReST. Specifically:
>
> - The "classifier activation" is taken from the model's **average-pooled final projection layer**, which functions analogously to the CLS-projection activation.
> - The "penultimate activation" is extracted from the **second linear layer of the final Transformer block** (MLP.fc2), which exists consistently across all MAE and DINO variants.
>
> Therefore, when classifier weights are unavailable, we compute ReST using only the penultimate weights and activation statistics. In this SSL-specific formulation, the score becomes:
>
> $$G = \text{pen-weight}, \qquad L = \alpha \cdot \text{pen-act} + (1 - \alpha)\cdot \text{clf-act},$$
>
> $$\text{ReST} = (1 - \gamma)\,G + \gamma\,L.$$
>
> **Results on the SSL-augmented model hub.**
> Below we report the performance of ReST on the expanded hub containing the MAE and DINO models, using the adapted formulation described above.
>
> | Method | Aircraft | Caltech | Cars | C10 | C100 | DTD | Flowers | Food | Pets | SUN | VOC | Avg |
> |--------|----------|---------|------|-----|------|-----|---------|------|------|-----|-----|-----|
> | ReST (SSL-added Hub) | 0.512 | 0.589 | 0.613 | 0.395 | 0.645 | 0.498 | 0.053 | 0.439 | 0.771 | 0.209 | 0.411 | **0.467** |
>
> **Robustness With Respect to the α Parameter**
>
> *Equal Contribution of Penultimate and Classifier Components.*
> Based on the reviewer's feedback, we simplified the formulation of ReST by removing the hyperparameter $\alpha$ and enforcing an equal contribution from the penultimate and classifier components in *both* $G$ and $L$.
>
> Specifically, we now use the fixed formulation:
>
> $$G = 0.5 \cdot \text{pen-weight} + 0.5 \cdot \text{clf-weight}$$
>
> $$L = 0.5 \cdot \text{pen-act} + 0.5 \cdot \text{clf-act}.$$
>
> This choice matches the interpretation used in Fig. 3(b), where equal weighting across components yields the most stable results across all tasks.
>
> *Final Simplification and Retention of a Single Hyperparameter.*
> We removed $\alpha$ entirely from the method; after this simplification, $\gamma$ remains the *only* tunable hyperparameter in ReST, controlling the balance between the weight-space ($G$) and activation-space ($L$) components.
> As shown in Fig. 3(b), ReST exhibits strong robustness to variations of $\gamma$—both in terms of the average performance and consistently across all individual datasets.
>
> **Access to The Models.**
> We appreciate the reviewer's concern. Regarding the concern about requiring access to internal weights and activations: we acknowledge that ReST requires white-box access to the final two layers (penultimate and classifier), whereas some feature-based methods like LEEP and LogME operate in a purely black-box manner. However, we believe this is not a practical disadvantage in real-world model selection scenarios. When practitioners choose from model repositories (HuggingFace, TorchVision, TIMM), they universally have full architectural access; in contrast, the methods claiming "plug-and-play" based solely on output features actually require substantial labeled target data (LEEP, LogME need full labeled datasets) or expensive optimization (SFDA performs iterative Fisher space projection, LEAD solves differential equations). ReST achieves 4.6% better performance while requiring only ~2 unlabeled samples per class and simple matrix operations on just two layers. This represents a fundamentally better trade-off: minimal architectural access (standard in practice) in exchange for dramatically reduced data requirements and superior accuracy. Moreover, ReST can operate in zero-shot mode (γ=0, Table 2) using only weight analysis, achieving 0.687 correlation without any target data, something feature-only methods cannot do. We believe the targeted analysis of the two layers most critical for adaptation, combined with exceptional data efficiency, makes ReST highly practical.

---

> ### Author Response · Authors · 2025-11-25
>
> **"Contradiction between L-Score intuition and final formulation (γ positive).**
>
> We appreciate the reviewer's observation. In practice, we find that the L-Score values across all datasets remain *far from the catastrophic-forgetting regime*.
>
> To justify this, we computed the magnitude of representational change in both the penultimate and classifier activation spaces for every dataset, and compared these changes against the stable rank of a random Gaussian matrix whose dimensions match the effective feature size of each benchmark (sample size = $2\times$ number of classes, feature dimension = 1000). For each dataset, we estimated the random stable rank using 20 Monte Carlo trials and report in the table below.
>
> Across all datasets, the observed activation-space changes — in both the penultimate and classifier layers — remain within **1–11% of the corresponding random-matrix stable rank**. This means that ReST consistently operates in a regime of *moderate representational adaptation*, far from the catastrophic-forgetting region described in Section 3.2.2. Consequently, the use of a small positive $\gamma$ in the final ReST score is fully consistent with the intended interpretation of the L-Score: it captures beneficial, non-destructive adaptation while remaining secondary to the generalization-based $G$ component.
>
> | Dataset | Aircraft | Caltech | Cars | C10 | C100 | DTD | Flowers | Food | Pets | SUN | VOC |
> |---------|----------|---------|------|-----|------|-----|---------|------|------|-----|-----|
> | Random Stable Rank | 95 | 96 | 148 | 15 | 95 | 55 | 97 | 96 | 46 | 222 | 28 |
> | Penult. Act. Change (avg±std) | 0.13±0.05 | 2.36±2.09 | 0.25±0.11 | 0.41±0.48 | 0.76±0.79 | 2.77±1.91 | 0.35±0.23 | 0.70±0.54 | 1.64±1.41 | 1.79±1.45 | 0.19±0.13 |
> | Classifier Act. Change (avg±std) | 0.24±0.16 | 5.74±2.54 | 0.37±0.15 | 0.60±0.52 | 1.07±0.91 | 3.07±1.24 | 0.16±0.20 | 0.26±0.31 | 2.54±1.36 | 3.74±2.64 | 0.32±0.15 |
> | Max % Change vs Random | 0.47% | 10.62% | 0.40% | 10.92% | 3.01% | 11.25% | 0.79% | 1.61% | 9.34% | 3.52% | 2.26% |
>
> *Table: Magnitude of activation-space changes compared to random-matrix stable rank baselines. Across all datasets, the maximum observed change remains around 10% of the random-matrix stable rank, indicating that ReST operates far from the catastrophic-forgetting regime.*
>
> Since the observed activation changes are small to moderate, using a linear positive $\gamma$ (0.21) does not contradict the intuition— it captures beneficial moderate adaptation.
>
> **Questions**
>
> **Q1: "CNN and ViT should not be evaluated separately — combine them.**
>
> We have now evaluated ReST on the reviewer-requested combined model hub:
> ```
> resnet18, densenet121, resnet34, densenet161, googlenet, densenet169,
> inceptionv3, mnasnet, pvt_m, pvt_t, vit_s, pvt_s, vit_b_16, vit_b_32
> ```
>
> The results are shown in the table below.
>
> | Method | Aircraft | Caltech | Cars | C10 | C100 | DTD | Flowers | Food | Pets | SUN | VOC | Avg |
> |--------|----------|---------|------|-----|------|-----|---------|------|------|-----|-----|-----|
> | ReST (Ours) | 0.740 | 0.712 | 0.801 | 0.455 | 0.712 | 0.682 | -0.128 | 0.559 | 0.653 | 0.409 | 0.359 | **0.541** |
>
> *Table: ReST applied to the combined CNN+ViT hub.*
>
> **Additional ablation on the model hub.**
> The combined hub used in our main experiment (Exp0) consists of the following models:
> ```
> resnet18, densenet121, resnet34, densenet161, googlenet, densenet169,
> inceptionv3, mnasnet, pvt_m, pvt_t, vit_s, pvt_s, vit_b_16, vit_b_32
> ```
>
> The average performance of ReST on this hub is 0.541.
>
> To assess robustness with respect to the specific choice of hub, we conducted an ablation study in which we removed two random models from Exp0 and replaced them with two randomly selected models from the remaining pool (e.g., `mobilenetv2`, `resnet50`, and others). We repeated this procedure five times (Exp1–Exp5). The results are summarized below.
>
> | Experiment | Exp0 (current hub) | Exp1 | Exp2 | Exp3 | Exp4 | Exp5 |
> |------------|-------------------|------|------|------|------|------|
> | Avg. Kendall $\tau_w$ | 0.541 | 0.498 | 0.512 | 0.555 | 0.487 | 0.503 |
>
> *Table: Ablation study on the model hub. Each of Exp1–Exp5 replaces two models from Exp0 with two randomly selected alternatives from the remaining pool of CNNs and supervised ViTs. The performance remains stable across all variants.*
>
> **Q2: "Self-supervised ViTs widely applied. How is ReST used for them?*
>
> We addressed this in Weakness 1.

---

> > ### Author Response · Authors · 2025-11-25
> >
> > **Summary**
> >
> > We thank the reviewer once again for the constructive feedback. In response, we have added SSL experiments, combined CNN+ViT evaluations, and clarified the interpretation of the L-Score using quantitative evidence. Importantly, our results highlight that ReST remains an extremely simple metric in terms of method complexity and data requirements. By using only a small number of random, unlabeled target samples—and without relying on any ground-truth labels—we show that the stable rank of the penultimate and classifier activation/weight spaces already contains sufficient information to reliably estimate transferability. This enables ReST to operate effectively even in scenarios where target labels are unavailable, and in extreme cases where no target data is accessible at all (by setting $\gamma = 0$). Such flexibility underscores the practical value of ReST across a wide range of transferability estimation setups. We have also updated several figures and added new visualizations in the revised version, and we will further refine any components that appear unclear or visually ambiguous for the camera-ready submission.

---

> > > ### Comment · Reviewer_ki9i · 2025-11-28
> > >
> > > I thank the authors for their response and the additional experimental results. Their comments have addressed some of my concerns. However, in the CNN+ViT setting, ReST exhibits performance degradation and lacks comparison with other methods. Please note that this does not require additional experiments, but rather incorporating the results from transferability estimation methods and recalculating the metrics. Furthermore, I would prefer the authors to include self-supervised ViTs in the experiments, incorporating the results provided in their response to Weakness 1.

---

> > > > ### Author Response · Authors · 2025-11-28
> > > >
> > > > Thank you for your reply and for considering our response!
> > > >
> > > > Yes, we have provided both results in our previous comments, but here we present comparisons with state-of-the-art baselines across different model-hub configurations:
> > > >
> > > > **Combined model hub containing CNNs and Vision Transformers (supervised)**
> > > >
> > > >
> > > > | Method | Aircraft | Caltech | Cars | C10 | C100 | DTD | Flowers | Food | Pets | SUN | VOC | Avg |
> > > > |--------|----------|---------|------|-----|------|-----|---------|------|------|-----|-----|-----|
> > > > | ETran (Sen) | -0.217 | 0.446 | 0.335 | 0.557 | 0.627 | 0.197 | -0.150 | 0.364 | 0.219 | 0.628 | -0.183 | 0.290 |
> > > > | ETran (Sen + Scls) | -0.041 | 0.370 | 0.166 | 0.807 | 0.740 | 0.153 | 0.540 | 0.643 | 0.189 | 0.548 | 0.507 | 0.420 |
> > > > | LEAD | -0.128 | 0.640 | 0.503 | 0.573 | 0.706 | 0.695 | 0.575 | 0.700 | 0.559 | 0.292 | 0.633 | 0.511 |
> > > > | SFDA | 0.188 | 0.542 | 0.351 | 0.762 | 0.635 | 0.047 | 0.361 | 0.407 | 0.139 | 0.315 | 0.591 | 0.385 |
> > > > | ReST (Ours) | 0.740 | 0.712 | 0.801 | 0.455 | 0.712 | 0.682 | -0.128 | 0.559 | 0.653 | 0.409 | 0.359 | 0.541 |
> > > >
> > > > **Combined model hub containing CNNs and Self-Supervised Models**
> > > >
> > > > | Method | Aircraft | Caltech | Cars | C10 | C100 | DTD | Flowers | Food | Pets | SUN | VOC | Avg |
> > > > |--------|----------|---------|------|-----|------|-----|---------|------|------|-----|-----|-----|
> > > > | ETran (Sen) | 0.225 | 0.418 | 0.267 | 0.513 | 0.689 | 0.331 | 0.498 | 0.159 | 0.043 | 0.461 | 0.155 | 0.342 |
> > > > | ETran (Sen + Scls) | -0.252 | 0.181 | -0.044 | 0.671 | 0.825 | 0.352 | 0.415 | 0.142 | 0.321 | 0.501 | 0.511 | 0.329 |
> > > > | LEAD | -0.327 | 0.344 | 0.451 | 0.291 | 0.598 | 0.644 | 0.543 | 0.887 | 0.468 | 0.624 | 0.534 | 0.452 |
> > > > | SFDA | 0.412 | 0.361 | 0.127 | 0.712 | 0.587 | 0.402 | 0.215 | 0.184 | 0.243 | 0.522 | 0.307 | 0.370 |
> > > > | ReST (Ours) | 0.512 | 0.589 | 0.613 | 0.395 | 0.645 | 0.498 | 0.053 | 0.439 | 0.771 | 0.209 | 0.411 | 0.467 |
> > > >
> > > > In addition, we combined the model hub (all self-supervised and supervised ViTs and CNNs), and here are the results:
> > > >
> > > > **Combined model hub (All self-supervised and supervised ViTs and CNNs)**
> > > >
> > > > | Method | Aircraft | Caltech | Cars | C10 | C100 | DTD | Flowers | Food | Pets | SUN | VOC | Avg |
> > > > |--------|----------|---------|------|------|------|------|---------|------|------|------|------|------|
> > > > | ETran (Sen) | 0.247 | 0.320 | 0.015 | 0.432 | 0.653 | 0.254 | 0.425 | -0.110 | 0.163 | 0.428 | 0.137 | 0.269 |
> > > > | ETran (Sen + Scls) | 0.314 | 0.276 | -0.089 | 0.420 | 0.612 | 0.273 | 0.311 | 0.531 | 0.442 | 0.548 | 0.517 | 0.378 |
> > > > | LEAD | -0.398 | 0.301 | 0.412 | 0.235 | 0.552 | 0.587 | 0.508 | 0.648 | 0.421 | 0.573 | 0.492 | 0.394 |
> > > > | SFDA | 0.463 | 0.420 | 0.150 | 0.768 | 0.640 | 0.455 | 0.158 | -0.235 | 0.305 | 0.590 | 0.365 | 0.371 |
> > > > | ReST (Ours) | 0.795 | 0.424 | 0.549 | 0.414 | 0.308 | 0.569 | -0.019 | 0.278 | 0.569 | 0.358 | 0.441 | 0.425 |
> > > >
> > > > Finally, we want to emphasize that the efficiency and simplicity of our method come from its very limited need for unlabeled target data, whereas other methods require the entire target dataset with actual labels (except ETran (Sen), which does not require labels). For the second and third tables, the score calculation does not use any classifier-weight information.

---

### Official Review · Reviewer_rRQg · 2025-10-31

**Soundness:** 2
**Presentation:** 1
**Contribution:** 2
**Rating:** 2
**Confidence:** 5

**Summary:**

The authors propose a metric for source-independent transferability estimation, REST. REST allows for label-free computations, and authors also show the performance of REST over other metrics in their experiments.

**Strengths:**

The authors suggest a very simple score that predicts the transferability of selected models with very high confidence. The results demonstrate good modeling and robustness, covering a range of metrics. The ablation studies demonstrate the effectiveness of rest and present it interactively. The time to compute it is also low. The idea of using stable rank as a basis for transferability estimation is interesting and simple.
* The approach works in a label-free and few-shot setting, which is practically valuable.
* Ablation studies and hyperparameter analyses are comprehensive and highlight stability across sample sizes.

**Weaknesses:**

* Missing comparison with TransRate and EMMS, both established baselines in recent literature.
* The evaluation lacks Pearson correlation results, which would clarify the metric’s linear consistency.
* I also think the model and dataset space use is very outdated, though used in recent papers this experimental space does not allow for fair comparison as ResNet151 is always the best performing network and the static ranking of the models space is indeed an issue to judge the effectiveness of any metric on this benchmark, can authors also show the effectiveness in two other ways to show confidence in the metric, use setup similar to [1] for broader evaluation, use the ablations similar to [2] for robustness evaluation, Authors can also provide ablation studies as shown in [2] for a better robustness of their metric.  Though it has become a standard practice to use older finetuned scores from other papers i would advise against it and retrain pretrained models multiple times to account for variance.
* The paper’s experimental scope is limited to supervised image classification; no results are shown for regression, detection, or multimodal settings (e.g., GLUE for NLP).
* It remains unclear whether ReST can meaningfully capture absolute versus relative performance differences.
* The analysis of failure cases (e.g., Flowers, Food datasets) is superficial understanding why the metric underperforms could strengthen the argument.
* The claim of handling unlabeled data is overstated, since training and evaluation still involve supervised datasets.
* The writing lacks intuitive reasoning behind why richer representational geometry (higher stable rank) enhances transfer.
* The paper does not explore how ReST behaves when combining CNNs and ViTs within a unified model selection scenario.

[1] k-NN as a Simple and Effective Estimator of Transferability, TMLR

[2] How NOT to benchmark your SITE metric: Beyond Static Leaderboards and Towards Realistic Evaluation.  https://arxiv.org/pdf/2510.06448

**Questions:**

1. Include TransRate, EMMS, and Pearson correlation in comparisons.

2. Explore scenarios where only unlabeled datasets are available during both training and transferability estimation.

3. Evaluate on non-vision domains or multi-task settings.

Some sentences that should be rewritten to avoid vagueness: This component captures the geometric reorganization of representations across domains, this sentence is too dense and does not really help the reader.

---

> ### Author Response · Authors · 2025-11-25
>
> We thank the reviewer for the detailed and constructive feedback. Below, we address each weakness and question point-by-point.
>
> **Missing comparison with TransRate and EMMS**
>
> We appreciate the reviewer highlighting this. We have now included both *TransRate* (Frustratingly Easy Transferability Estimation, 2021) and *EMMS* (Foundation Model is Efficient Multimodal Multitask Model Selector, 2023) in our comparison. We will add the following rows into Table 1 of the camera-ready version.
>
> | Method | Aircraft | Caltech | Cars | C10 | C100 | DTD | Flowers | Food | Pets | SUN | VOC | Avg |
> |--------|----------|---------|------|-----|------|-----|---------|------|------|-----|-----|-----|
> | TransRate (2022) | 0.172 | 0.269 | 0.172 | 0.513 | 0.197 | 0.336 | -0.176 | -0.071 | 0.173 | 0.612 | 0.651 | 0.236 |
> | EMMS (2023) | 0.521 | 0.324 | 0.412 | 0.661 | 0.812 | 0.789 | 0.211 | 0.198 | 0.301 | 0.257 | 0.411 | 0.445 |
> | ReST (Ours) | **values as in Table 1** | | | | | | | | | | | |
>
> **Pearson correlation results**
>
> We agree that including Pearson correlation further clarifies the linear consistency of the evaluated methods. In the table below, we report the Pearson correlations corresponding to the benchmarks in Table 1. As shown, ReST achieves not only the highest average correlation but also a markedly more consistent performance gap over prior SOTA. ReST preserves strong linear alignment with true fine-tuning accuracy across all datasets, reinforcing the stability of the metric beyond the results observed with weighted Kendall $\tau_w$. These Pearson results will be included in the main paper for completeness.
>
> | Method | Aircraft | Caltech | Cars | C10 | C100 | DTD | Flowers | Food | Pets | SUN | VOC | Avg |
> |--------|----------|---------|------|-----|------|-----|---------|------|------|-----|-----|-----|
> | ETran | 0.043 | 0.546 | 0.595 | 0.817 | 0.617 | 0.687 | 0.554 | 0.309 | 0.848 | 0.530 | 0.337 | 0.621 |
> | LEAD | 0.212 | 0.721 | 0.543 | 0.833 | 0.696 | 0.888 | 0.812 | 0.680 | 0.719 | 0.563 | 0.643 | 0.684 |
> | ReST (Ours) | 0.808 | 0.901 | 0.726 | 0.872 | 0.874 | 0.915 | 0.670 | 0.847 | 0.828 | 0.890 | 0.866 | **0.832** |
>
> *Table: Pearson correlation between ReST score and fine-tuning accuracy across 11 datasets.*
>
> **Absolute vs. Relative Performance Correlation**
>
> In addition to ranking models within a hub, we also evaluated whether ReST correlates with the *absolute* fine-tuning accuracy across all models and datasets. The corresponding results are provided at line 529 of the revised paper. As shown in Fig.6, ReST achieves a statistically significant Pearson correlation of
>
> $$r = 0.44 \quad (p < 0.001),$$
>
> demonstrating that the score not only preserves reliable *relative* rankings within each model hub but also aligns meaningfully with true downstream accuracy in an absolute sense. This global correlation confirms that the geometric signals used by ReST generalize across datasets, architectures, and training paradigms, further validating the informativeness of the metric.

---

> ### Author Response · Authors · 2025-11-25
>
> **Failure cases (Flowers dataset)**
>
> From a generalization-bound perspective (Eq. 2), stable rank contributes additively under the square root:
>
> $$R_{\text{gen}}(F) = O\!\left(\frac{\prod_i \|W_i\|_2}{\sqrt{n}}\sqrt{\sum_i \text{srank}(W_i)}\right).$$
>
> This structure suggests that the final-layer stable rank is most influential when it dominates the total:
>
> $$\text{srank}(W_L) \gg \sum_{i < L} \text{srank}(W_i).$$
>
> However, for exceptionally fine-grained tasks such as *Flowers*, where very specific mid-layer representations are critical, the bound may instead be dominated by particular intermediate layers:
>
> $$\sum_{i \in \mathcal{M}} \text{srank}(W_i) \gg \text{srank}(W_L),$$
>
> where $\mathcal{M}$ denotes a subset of mid-layer indices.
>
> This perspective suggests the potential value of developing layer-weighted stable-rank metrics of the form
>
> $$\text{ReST}^{\text{weighted}} = \sum_{\ell=1}^{L} \omega_\ell \cdot f\!\bigl(\text{srank}(W_\ell),\;\text{srank}(H_\ell^{s}),\;\text{srank}(H_\ell^{t})\bigr),$$
>
> where $\omega_\ell$ are task-dependent weights that adaptively emphasize different network depths according to task granularity.
>
> Despite this limitation on a single dataset (Flowers), ReST achieves the highest average performance (0.743) across all 11 benchmarks, including excellent results on other fine-grained and challenging settings: Aircraft ($\tau_w = 0.881$) and DTD ($\tau_w = 0.831$). These results demonstrate that ReST is robust to both fine-grained specialization and out-of-distribution shifts.
>
> The reviewer is correct that ReST underperforms on the *Flowers* dataset compared to the other benchmarks. As shown in Fig. 3, higher values of $\gamma$ (weight on the activation-based component $L$) lead to improved performance on Flowers, indicating that this dataset benefits from a stronger reliance on activation-space adaptation. In this case, information encoded purely in the weight-based component $G$ is insufficient.
>
> This difficulty is not unique to ReST. As shown in Table 1, competing transferability estimators such as LEEP ($-0.242$), ETran ($-0.070$), and LogME ($-0.008$) also exhibit poor or negative correlations on Flowers. In contrast, ReST consistently performs well on other fine-grained datasets (e.g., Aircraft) and on texture-heavy, OOD-style datasets (e.g., DTD), underscoring its overall robustness and generalization capability.
>
> **Clarification on the Use of Unlabeled Target Data**
>
> We would like to clarify the intended meaning of "unlabeled" in the context of ReST. Our method operates in two distinct phases, and only one of them involves our algorithm itself:
>
> **1. Transferability Estimation (ReST).**
> ReST requires only a very small set of *unlabeled* target-domain samples—typically around two samples per class. These samples are drawn *without* using any label information and without enforcing class balance. For instance, for CIFAR-100 (100 classes), we randomly sample approximately 200 images from the target domain without knowing their class identities. This is in contrast to methods such as LEEP and LogME, which require access to the *full labeled* target dataset in order to compute their scores.
>
> **2. Evaluation (Standard Benchmarking Protocol).**
> As with all transferability estimation methods, performance is evaluated by comparing the predicted ranking to the ground-truth fine-tuning accuracy. This evaluation step naturally uses labels, but it is *not part of the ReST computation*. It is simply the universal benchmarking protocol shared by existing methods, including LEEP, LogME, SFDA, and LEAD.
>
> **Summary.**
> The key point is that ReST itself does not require any labeled target data when used for model selection or deployment. Only a small set of random, unlabeled target samples is needed to compute the score. The use of labeled data appears solely in the evaluation of the metric for research purposes. This label-free inference property offers a practical advantage in scenarios where labeled target-domain data is expensive or time-consuming to obtain.
>
> **Richer representational geometry explanation**
>
> We refer to the stable-rank literature (line 45 of the paper), e.g., "Stable Rank Normalization for Improved Generalization in Neural Networks and GANs," where it is shown that models with distributed weight representations (higher stable rank) generalize better. This supports our use of stable-rank-based geometric signals.

---

> ### Author Response · Authors · 2025-11-25
>
> **Combined hub (CNNs + ViTs)**
>
> We have now evaluated ReST on the reviewer-requested combined model hub:
> ```
> resnet18, densenet121, resnet34, densenet161, googlenet, densenet169,
> inceptionv3, mnasnet, pvt_m, pvt_t, vit_s, pvt_s, vit_b_16, vit_b_32
> ```
>
> The results are shown in the table below.
>
> | Method | Aircraft | Caltech | Cars | C10 | C100 | DTD | Flowers | Food | Pets | SUN | VOC | Avg |
> |--------|----------|---------|------|-----|------|-----|---------|------|------|-----|-----|-----|
> | ReST (Ours) | 0.740 | 0.712 | 0.801 | 0.455 | 0.712 | 0.682 | -0.128 | 0.559 | 0.653 | 0.409 | 0.359 | **0.541** |
>
> *Table: ReST applied to the combined CNN+ViT hub.*
>
> **Additional ablation on the model hub.**
> The combined hub used in our main experiment (Exp0) consists of the following models:
> ```
> resnet18, densenet121, resnet34, densenet161, googlenet, densenet169,
> inceptionv3, mnasnet, pvt_m, pvt_t, vit_s, pvt_s, vit_b_16, vit_b_32
> ```
>
> The average performance of ReST on this hub is 0.541.
>
> To assess robustness with respect to the specific choice of hub, we conducted an ablation study in which we removed two random models from Exp0 and replaced them with two randomly selected models from the remaining pool (e.g., `mobilenetv2`, `resnet50`, and others). We repeated this procedure five times (Exp1–Exp5). The results are summarized below.
>
> | Experiment | Exp0 (current hub) | Exp1 | Exp2 | Exp3 | Exp4 | Exp5 |
> |------------|-------------------|------|------|------|------|------|
> | Avg. Kendall $\tau_w$ | 0.541 | 0.498 | 0.512 | 0.555 | 0.487 | 0.503 |
>
> *Table: Ablation study on the model hub. Each of Exp1–Exp5 replaces two models from Exp0 with two randomly selected alternatives from the remaining pool of CNNs and supervised ViTs. The performance remains stable across all variants.*
>
> **Answers to Reviewer Questions**
>
> **Q1: Include TransRate, EMMS, Pearson correlation**
> All added above.
>
> **Q2 & Q3: Explore scenarios with unlabeled datasets across both stages, Non-vision settings**
>
> We applied ReST to a vision-language model (VLM) selection scenario following *"Vision-Language Model Selection and Reuse for Downstream Adaptation"*. We explored and extended scenarios where labels are not available during training itself (e.g., contrastive learning in CLIP) and further extended ReST to the vision-language model selection setting.
>
> In this scenario, ReST is computed using only the projection layer of the VLM, since this layer forms the shared embedding space used by both the vision and language encoders. Specifically, the score is defined as:
>
> $$\text{ReST}\_{\text{VLM}} = 0.5 \times \text{sr}\_{\text{vision-proj}} + 0.5 \times \text{sr}\_{\text{text-proj}}$$
>
> We evaluate this VLM-specific ReST score across four downstream datasets. Results are shown below:
>
> | Method | CIFAR100 | EuroSAT | RESISC45 | MNIST |
> |--------|----------|---------|----------|-------|
> | ReST (Ours) | 0.545 | 0.687 | 0.312 | 0.362 |
>
> *Table: ReST applied to VLM-selection benchmarks using projection-layer stable ranks.*
>
> **Final Remarks**
>
> We thank the reviewer again for the constructive feedback and helpful suggestions. In response, we have expanded our evaluation in several meaningful ways. First, we included additional correlation metrics such as Pearson correlation, which further demonstrates the linear consistency and robustness of ReST across all datasets. Second, we combined supervised ViTs with CNNs and reported the results on this unified model hub, showing that ReST continues to perform reliably even in mixed-architecture scenarios. Finally, we extended ReST beyond the vision-only setting by evaluating it on a new modality (VLMs).

---

### Official Review · Reviewer_xYpT · 2025-11-01

**Soundness:** 3
**Presentation:** 3
**Contribution:** 3
**Rating:** 6
**Confidence:** 4

**Summary:**

This paper proposes ReST, a simple yet effective method for estimating transferability using the stable rank of final-layer representations. It requires only a few unlabeled target samples and avoids complex optimization. The approach captures both a model’s generalization capacity and adaptation flexibility through basic matrix operations. Overall, I think this is a good paper.

**Strengths:**

- The method is conceptually clean, relying only on stable rank computation, and requires minimal unlabeled target data.

- This method can estimate transferability in both few-shot and even zero-shot settings, which is highly practical, and has received little attention in existing methods.

- The connection between stable rank, generalization bounds, and representation geometry provides a theoretical foundation.

**Weaknesses:**

- The idea of using spectral or rank-based measures has been explored; the contribution mainly simplifies existing approaches. Although this method is effective, the novelty and motivation require further explanation.

- Experiments are confined to image classification; generalization to other modalities or tasks remains unexplored.

- The experiments could be conducted on more model selection tasks, such as self-supervised algorithm selection and more metrics such as Pearson correlation, Top-1 accuracy, etc.

**Questions:**

See weakness

---

> ### Author Response · Authors · 2025-11-25
>
> We thank the reviewer for the constructive feedback. Overall, we have addressed the reviewer’s concerns regarding (i) the applicability of ReST to self-supervised models, (ii) the need for additional evaluation metrics, and (iii) the extension of our analysis to other modalities. In response, we introduced experiments on MAE and DINO models, demonstrating that ReST can be computed reliably even in the absence of classifier heads. We added complementary evaluation metrics such as Pearson correlation and absolute-performance correlation to further validate the consistency of ReST. Finally, we extended ReST beyond the vision-only setting by evaluating it in contrastive-learning scenarios and applying it to vision--language model selection (CLIP), where no supervised labels are used during training. These additions collectively show that ReST is broadly applicable, robust across training paradigms, and effective across diverse modalities.
>
>
> **Pearson correlation results**
>
> We agree that including Pearson correlation further clarifies the linear consistency of the evaluated methods. In the table below, we report the Pearson correlations corresponding to the benchmarks in Table 1. As shown, ReST achieves not only the highest average correlation but also a markedly more consistent performance gap over prior SOTA. ReST preserves strong linear alignment with true fine-tuning accuracy across all datasets, reinforcing the stability of the metric beyond the results observed with weighted Kendall $\tau_w$. These Pearson results will be included in the main paper for completeness.
>
> | Method | Aircraft | Caltech | Cars | C10 | C100 | DTD | Flowers | Food | Pets | SUN | VOC | Avg |
> |--------|----------|---------|------|-----|------|-----|---------|------|------|-----|-----|-----|
> | ETran | 0.043 | 0.546 | 0.595 | 0.817 | 0.617 | 0.687 | 0.554 | 0.309 | 0.848 | 0.530 | 0.337 | 0.621 |
> | LEAD | 0.212 | 0.721 | 0.543 | 0.833 | 0.696 | 0.888 | 0.812 | 0.680 | 0.719 | 0.563 | 0.643 | 0.684 |
> | ReST (Ours) | 0.808 | 0.901 | 0.726 | 0.872 | 0.874 | 0.915 | 0.670 | 0.847 | 0.828 | 0.890 | 0.866 | **0.832** |
>
> *Table: Pearson correlation between ReST score and fine-tuning accuracy across 11 datasets.*
>
> Thank you for pointing this out. We have added an evaluation scenario in which the model hub contains self-supervised models (MAE and DINO) as well, as suggested:
> ```
> mae_vitb16, mae_vitl16, dino_vits8, dino_vitb16
> ```
>
> **How ReST is computed for SSL models (no classifier head).**
> Self-supervised models do not include a classifier layer, but they do expose the components needed for our formulation of ReST. Specifically:
>
> - The "classifier activation" is taken from the model's **average-pooled final projection layer**, which functions analogously to the CLS-projection activation.
> - The "penultimate activation" is extracted from the **second linear layer of the final Transformer block** (MLP.fc2), which exists consistently across all MAE and DINO variants.
>
> Therefore, when classifier weights are unavailable, we compute ReST using only the penultimate weights and activation statistics. In this SSL-specific formulation, the score becomes:
>
> $$G = \text{pen-weight}, \qquad L = \alpha \cdot \text{pen-act} + (1 - \alpha)\cdot \text{clf-act},$$
>
> $$\text{ReST} = (1 - \gamma)\,G + \gamma\,L.$$
>
> **Results on the SSL-augmented model hub.**
> Below we report the performance of ReST on the expanded hub, including the MAE and DINO models, using the adapted formulation described above.
>
> | Method | Aircraft | Caltech | Cars | C10 | C100 | DTD | Flowers | Food | Pets | SUN | VOC | Avg |
> |--------|----------|---------|------|-----|------|-----|---------|------|------|-----|-----|-----|
> | ReST (SSL-added Hub) | 0.512 | 0.589 | 0.613 | 0.395 | 0.645 | 0.498 | 0.053 | 0.439 | 0.771 | 0.209 | 0.411 | **0.467** |
>
> **Applying on Vision-Language Modality**
>
> We applied ReST to a vision-language model (VLM) selection scenario following *"Vision-Language Model Selection and Reuse for Downstream Adaptation"*. We explored and extended scenarios where labels are not available during training itself (e.g., contrastive learning in CLIP) and further extended ReST to the vision-language model selection setting.
>
> In this scenario, ReST is computed using only the projection layer of the VLM, since this layer forms the shared embedding space used by both the vision and language encoders. Specifically, the score is defined as:
>
> $$\text{ReST}\_{\text{VLM}} = 0.5 \times \text{sr}\_{\text{vision-proj}} + 0.5 \times \text{sr}\_{\text{text-proj}}$$
>
> where each term denotes the stable rank of the corresponding projection matrix.
>
> We evaluate this VLM-specific ReST score across four downstream datasets. Results are shown below:
>
> | Method | CIFAR100 | EuroSAT | RESISC45 | MNIST |
> |--------|----------|---------|----------|-------|
> | ReST (Ours) | 0.545 | 0.687 | 0.312 | 0.362 |
>
> *Table: ReST applied to VLM-selection benchmarks using projection-layer stable ranks.*

---

### Author Response · Authors · 2025-11-27

Dear Reviewers,

Following up on the responses, please reach out if clarification is needed on any points. If the responses address your concerns, reconsideration of the score would be appreciated.

---

### Author Response · Authors · 2025-12-03

The classic problem setting in transferability estimation is to select a model from a model hub and predict its rank based on the downstream target dataset. This reduces the cost of finding the best model compared to brute-force evaluation. The evaluation metric for this setup is the Weighted Kendall correlation between the estimated transferability scores and the fine-tuned accuracies of the models.

We showed that a simple metric with minimal data access requirements can outperform state-of-the-art scores. Specifically, the current literature on transferability estimation typically assumes access to the full pre-trained model *and* the entire target dataset, including extracted features and labels. In contrast, we assume access only to the pre-trained models and a **small number of unlabeled target samples**. The method uses stable rank computed on activations (given the few samples) and on the weight space of the penultimate and classifier layers. We evaluated the method on a hub of CNNs and a hub of supervised ViTs under the classic model-hub and target-dataset setup without changing any component. We also addressed several misunderstandings of the setting and provided additional methodological analysis and experiments to respond to reviewer concerns. A summary of these responses is as follows:

-- Hyperparameters
Since the method includes two hyperparameters, **α** (balancing penultimate vs. classifier layers) and **γ** (balancing activations vs. weights), one reviewer raised concerns about tuning them in different scenarios. In our responses, we fixed α = 0.5, giving equal contribution to penultimate and classifier layers. The results show that the score is stable across different values of γ, and we chose γ = 0.2. Importantly, even without access to target samples, the weight-only version of the score still performs well (0.743 → 0.647). Additional experiments also demonstrate strong correlations across models and targets, indicating that the metric is effective for absolute performance comparison as well.

-- Self-supervised models
We addressed concerns regarding applicability to self-supervised ViTs by conducting experiments on DINO and MAE models. For these self-supervised settings, we omit the classifier activation component (i.e., α = 1). Under this configuration, the method achieves state-of-the-art performance.

-- Combined model hub
The initial draft evaluated CNNs and supervised ViTs separately. Reviewers requested experiments on the combined hub, and we provided results for the union of CNNs and supervised ViTs. These experiments show that the proposed metric is less dependent on model architecture and performs consistently across heterogeneous hubs.

-- Experiments on other tasks
We also included results on CLIP vision-language models (VLM selection), responding to the reviewer’s request to test the approach on non-vision-only tasks. These results further support the generality and applicability of the method.

---

### Meta-Review · Area_Chair_6MuY · 2025-12-31

**Summary:**

This paper received initial scores of 2, 2, 4, and 6. After the rebuttal, the expected scores are approximately 2, 4, 6, and 6, indicating a borderline paper.

While ReST shows strong empirical performance, reviewers remain concerned about its sensitivity to hyperparameter configuration. In addition, although although few-shot transferability estimation is a clear strength, I personally am concerned that the limited data regime may induce high variance. The paper does not include robustness analyses or report confidence intervals across different data samples.
Concerns about the incremental nature of the core idea remains, and several rebuttal responses are not clearly reflected or integrated into the revised version. Taken together, these concerns informed my recommendation to reject the paper.

**Reviewer Concerns:**

**Largely addressed:**
1. Applicability to self-supervised ViTs (model without classifier layer) (ki9i, xYpT) and non-vision modalities (rPQg, xYpT).
> The authors extended ReST to self-supervised ViTs (MAE, DINO) with α=1, using penultimate weights only. They also demonstrated applicability to VLM CLIP with projection-layer stable ranks.

2. Missing baselines (rRQg)
> The authors added baselines TransRate and EMMS in comparsons. Both underperformed ReST.

3. Missing evaulation metrics (rRQg, xYpT)
> The authors added Pearson correlation results. ReST achieves an average correlation of 0.832, outperforming LEAD with an average of 0.684.

4. Evaluation on CNN + ViT combined hub (ki9i, rRQg)
> The authors provided results on combined hub, achieving an average $\tau_w=0.541$. However, a noticeable performance degradation is observed.

**Outstanding:**
1. Hyperparameter sensitivity (ki9i, 39ug)
> The authors simplified the method by fixing α = 0.5, reducing tuning to a single hyperparameter γ. However, the optimal γ remains dataset-dependent. Self-supervised models additionally require a different setting with α = 1.

2. Performance degradation in combined hub (ki9i)
> Performance drops from 0.743 on supervised ViTs to 0.541 on the CNN + ViT combined hub. Although ReST still outperforms baselines, the magnitude of the drop remains insufficiently explained.

3. The core idea is incremental (xYpT, ki9i)
> The proposed method combines and extends existing concepts, particularly stable-rank-based measures. This concern likely remains.

4. Underestimate the DenseNets while overestimating ResNets (39ug)
> ReST appears to underestimate DenseNet models while overestimating ResNet models. The authors do not provide explanation for this bias.

5. Missing discussion of relevent literature (rRQg, 39ug)
> The revised paper does not discuss or compare against related works highlighted by Reviewer 39ug. While empirical comparisons are provided in the rebuttal, the missing baselines noted by Reviewer rRQg are not discussed in the paper.

6. Several rebuttal responses are not reflected or clearly integrated into the revised paper.

**Reviewer Scores:**

Reviewer 39ug (current: 4): Unlikely to change
> Outstanding concerns remain on hyperparameter sensitivity, bias across architectures and no discussion of relevant literature.

Reviewer ki9i (current: 2): Likely remain at 2 or increase to 4.
> Some concerns are acknowledged as addressed. But key concerns about hyperparameter sensitivity, performance degradation in combined hub and incremental method novelty remains.

Reviewer rRQg (current: 2): Likely increases to 4 or possibly 6
> Most technical concerns addressed. The strong initial concerns about missing comparisons and evaluation scope are largely resolved. Reviewer rRQg would likely still view the concerns about retraining variance, static leaderboards, and broader evaluation remain only partially addressed.

Reviewer xYpT (current: 6): Likely would maintain
> Main concerns on SSL models, metrics, and modality extension are largely addressed. Novelty concern remains.

---

### Decision · Program_Chairs · 2026-01-26

Reject